Evolution of gremlin 2 in cetartiodactyl mammals: gene loss coincides with lack of upper jaw incisors in ruminants

Opazo Juan C. jopazo@gmail.com 1
Zavala Kattina 1
Krall Paola 2
Arias Rodrigo A. 3
1 Instituto de Ciencias Ambientales y Evolutivas, Universidad Austral de Chile , Valdivia , Chile
2 Unidad de Nefrología, Universidad Austral de Chile , Valdivia , Chile
3 Instituto de Producción Animal, Universidad Austral de Chile , Valdivia , Chile
Ringo John
Electronic publication date: 2017 Jan 26
Publication date: 2017
Volume: 5
Electronic Location ID: e2901
Received 2016 Oct 9; Accepted 2016 Dec 12
Copyright: ©2017 Opazo et al.
Copyright year: 2017
Copyright holder: Opazo et al.
License: This is an open access article distributed under the terms of the Creative Commons Attribution License, which permits unrestricted use, distribution, reproduction and adaptation in any medium and for any purpose provided that it is properly attributed. For attribution, the original author(s), title, publication source (PeerJ) and either DOI or URL of the article must be cited.
License URL: https://creativecommons.org/licenses/by/4.0/

Keywords: Gene loss, Tooth development, Differential gene retention, Gene duplication, Gene family evolution, Rate acceleration

Funding: Fondo Nacional de Desarrollo Científico y Tecnológico FONDECYT 1160627 This work was funded by a grant from the Fondo Nacional de Desarrollo Científico y Tecnológico (FONDECYT 1160627) to JCO. The funders had no role in study design, data collection and analysis, decision to publish, or preparation of the manuscript.

==============================
Understanding the processes that give rise to genomic variability in extant species is an active area of research within evolutionary biology. With the availability of whole genome sequences, it is possible to quantify different forms of variability such as variation in gene copy number, which has been described as an important source of genetic variability and in consequence of phenotypic variability. Most of the research on this topic has been focused on understanding the biological significance of gene duplication, and less attention has been given to the evolutionary role of gene loss. Gremlin 2 is a member of the DAN gene family and plays a significant role in tooth development by blocking the ligand-signaling pathway of BMP2 and BMP4. The goal of this study was to investigate the evolutionary history of gremlin 2 in cetartiodactyl mammals, a group that possesses highly divergent teeth morphology. Results from our analyses indicate that gremlin 2 has experienced a mixture of gene loss, gene duplication, and rate acceleration. Although the last common ancestor of cetartiodactyls possessed a single gene copy, pigs and camels are the only cetartiodactyl groups that have retained gremlin 2. According to the phyletic distribution of this gene and synteny analyses, we propose that gremlin 2 was lost in the common ancestor of ruminants and cetaceans between 56.3 and 63.5 million years ago as a product of a chromosomal rearrangement. Our analyses also indicate that the rate of evolution of gremlin 2 has been accelerated in the two groups that have retained this gene. Additionally, the lack of this gene could explain the high diversity of teeth among cetartiodactyl mammals; specifically, the presence of this gene could act as a biological constraint. Thus, our results support the notions that gene loss is a way to increase phenotypic diversity and that gremlin 2 is a dispensable gene, at least in cetartiodactyl mammals.

Introduction

One of the main goals of evolutionary biology is to understand the genetic basis of phenotypic diversity. To address this question, scientists have made efforts to identify genes that are linked to phenotypes and to explore the phenotypic consequences of genetic variability. With the availability of whole genome sequences, it has been possible to compare different forms of variability, and variation in gene copy number has been described as an important source of genetic variability. To date, most of the research on this topic has been focused towards understanding the biological significance of gene duplication, and less attention has been given to the evolutionary role of gene loss (Olson, 1999; Albalat & Cañestro, 2016). In the literature, there are examples of gene loss being associated with positive impacts on fitness. For example, the loss of the CCR5 gene in humans is associated with resistance to AIDS (Dean et al., 1996), and the loss of hair keratin genes in cetaceans is interpreted as an adaptation associated with the transition from terrestrial to aquatic life (Nery, Arroyo & Opazo, 2014; Yim et al., 2014). Thus, evolutionary studies of genes that possess a clear link to a given phenotype represent an opportunity to understand the phenotypic effects of gene loss and gene dispensability.

Gremlin 2, previously known as a protein related to Dan and Cerberus (PRDC), is a member of the DAN gene family, a group of extracellular bone morphogenetic protein (BMP) inhibitors, which was originally identified in a gene trap screen for developmentally significant genes (Minabe-Saegusa et al., 1998). Gremlin 2, as an antagonist of BMPs (Kattamuri et al., 2012), plays a role in several developmental processes including organogenesis, body patterning, and tissue differentiation. In embryonic stages, this gene is expressed in the reproductive, nervous, respiratory, musculoskeletal, and integumentary systems (Müller, Knapik & Hatzopoulos, 2006). Alternatively, during adulthood, it is a widely expressed gene found in high levels in ovaries, brain, and spleen (Sudo et al., 2004).

In the literature, it has been shown that gremlin 2 interacts with BMP2 and BMP4 by blocking their ligand-signaling pathway (Sudo et al., 2004). Human genetic studies have indicated that gremlin 2 variation can influence one’s susceptibility of having a common tooth malformation (Kantaputra et al., 2015). Mutational analysis in seven out of 263 patients with different dental anomalies has revealed the presence of mutations predicted to cause disease. Five patients of this study carried the same heterozygous mutations (Ala13Val) while the other two were carriers of two different heterozygous missense mutations (Gln76Glu and Glu136Asp) (Kantaputra et al., 2015). This genetic study supports the notion that inheritance of hypodontia is autosomal dominant, and this is related to gremlin 2. Despite this, the study also gives evidence of incomplete penetrance and variable expressivity. Genetic experiments provide further support for the role of gremlin 2 in tooth development (Brommage et al., 2014); it has been shown that gremlin 2 deficient mice have upper and lower incisor teeth with markedly reduced breadth and depth, and the upper incisors are more severely affected than lower ones (Vogel et al., 2015). According to Vogel et al. (2015) no other significant phenotypic effects have been observed in grem2−∕− individuals, indicating that this gene could be dispensable. From a developmental perspective, it has been shown that the pathway that controls tooth differentiation is conserved in most mammals other than cetaceans, xenarthrans, and phocid seals (Armfield et al., 2013). In dolphins, it has been shown that expression of BMP4, which is one of the main targets of gremlin 2 (Sudo et al., 2004), is extended to the caudal region of the developing jaw, a region where the fibroblast growth factor 8 gene (FGF8) is express in most mammals (Armfield et al., 2013). This developmental difference could be related to the divergent dental phenotype of cetaceans. Similar results have been found during epibranchial placode development (Kriebitz et al., 2009). Within the same mammalian clade, other groups also have different dental morphologies. For example, ruminants do not possess incisors in the upper jaw; instead, they possess a dental pad. Canines are also absent in most ruminant species with the exception of elk and red deer. This particular dental phenotype has consequences in the way these animals process food, which is different compared to related species (herbivores) that possess incisors in the upper jaw (e.g.,  horse). Finally, pangolins and baleen whales, both edentulous (i.e., toothless) groups, represent the most extreme cases of dental modification in this group of mammals. The lack of teeth in these groups has been related to the inactivation of tooth-specific genes (e.g.,  C4orf26; Springer et al., 2016).

The main goal of this study was to investigate the evolutionary history of gremlin 2, a gene that plays a significant role in the tooth development, in cetartiodactyl mammals a group that possesses divergent tooth morphologies. Results from our analyses show that gremlin 2 has experienced a mixture of gene loss, gene duplication, and rate acceleration. Although the last common ancestor of cetartiodactyls possessed a single gene copy, pigs and camels are the only cetartiodactyl groups that have retained gremlin 2. According to the phyletic distribution of this gene and synteny analyses, we propose that gremlin 2 was lost in the common ancestor of ruminants and cetaceans between 56.3 and 63.5 million years ago as a product of a chromosomal rearrangement. Our analyses also indicate that the rate of evolution of gremlin 2 in pigs and camels has been accelerated, and the possession of gremlin 2 clearly differentiates these groups from all other cetartiodactyl mammals.

Materials and Methods

DNA data collection and phylogenetic analyses

We annotated gremlin 2 genes in representative species of laurasiatherian mammmals. Our study included representative species from the orders Carnivora: cat (Felis catus), Siberian tiger (Panthera tigris), dog (Canis familiaris), ferret (Mustela putorius), Weddell seal (Leptonychotes weddellii), Pacific walrus (Odobenus rosmarus), panda (Ailuropoda melanoleuca); Perissodactyla: Przewalsk’s horse (Equus ferus), horse (Equus caballus), donkey (Equus asinus), Eulipotyphla: European hedgehog (Erinaceus europaeus); Chiroptera: Black flying fox (Pteropus alecto), Large flying fox (Pteropus vampyrus), Egyptian fruit bat (Rousettus aegyptiacus); Cetartiodactyla: pig (Sus Scrofa), alpaca (Vicugna pacos), dromedary (Camelus dromedarius) and Bactrian camel (Camelus bactrianus); and Pholidota (Manis javanica) (Table S1). Mouse and kangaroo rat sequences were used as outgroups. Amino acid sequences were aligned using the L-INS-i strategy from MAFFT v.6 (Katoh & Standley, 2013). Nucleotide alignment was generated using the amino acid alignment as a template using the software PAL2NAL (Suyama, Torrents & Bork, 2006). Phylogenetic relationships were estimated using maximum likelihood and Bayesian approaches. We used the propose model tool of IQ-Tree (Trifinopoulos et al., 2016) to select the best-fitting models for each codon position (TVMe+I, K2P+G4 and HKY+G4, for first, second and third codon positions, respectively). We performed a maximumlikelihood analysis to obtain the best tree using the program IQ-Tree (Trifinopoulos et al., 2016); and nodes support was assessed with 1,000 bootstrap pseudoreplicates using the ultrafast routine. Bayesian searches were conducted in MrBayes v.3.1.2 (Ronquist & Huelsenbeck, 2003); two independent runs of six simultaneous chains for 20 × 106 generations were set, and every 2,500 generations were sampled using default priors. The run was considered to have reached convergence once the likelihood scores reached an asymptote and the average standard deviation of the split frequencies remained <0.01. We discarded all trees that were sampled before convergence, and we evaluated support for the nodes and parameter estimates from a majority rule consensus of the last 4,000 trees.

Assessments of conserved synteny

We examined genes found up- and downstream of gremlin 2 in the laurasiatherian mammal representative species. Synteny analyses were conducted for dog (Canis familiaris), panda (Ailuropoda melanoleuca), horse (Equus caballus), donkey (Equus asinus), European hedgehog (Erinaceus europaeus), Large flying fox (Pteropus vampyrus), Egyptian fruit bat (Rousettus aegyptiacus), alpaca (Vicugna pacos), dromedary (Camelus dromedarius), pig (Sus scrofa), sheep (Ovis aries), goat (Capra hircus), cow (Bos taurus), minke whale (Balaenoptera acutorostrata), killer whale (Orcinus orca), baiji (Lipotes vexillifer), and Malayan pangolin (Manis javanica). Initial ortholog predictions were derived from the EnsemblCompara database (Herrero et al., 2016) and were visualized using the program Genomicus v85.01 (Muffato et al., 2010). In other cases, the genome data viewer platform from the National Center for Biotechnology information was used.

Results and Discussion

Gremlin 2 is a protein-coding gene located on the reverse strand that has two exons, where the first is the one that possesses the information for the final protein. In most laurasitherian mammals and humans the length of the coding portion of the gene (507 bp) is well conserved while some variation is found in cetartiodactyl mammals.

Figure 1 Maximum likelihood phylogenetic tree depicting relationships among gremlin 2 genes in laurasiatherian mammals.

Numbers on the nodes correspond to Bayesian posterior probabilities and maximum likelihood bootstrap support values. Sequences of mouse and kangaroo rat were used as outgroups.

Phylogenetic relationships

We constructed a phylogenetic tree in which we included representative species of laurasiatherian mammals (Fig. 1). Our phylogenetic analysis recovered the monophyly of each laurasiatherian order included in our sampling (Fig. 1). Although the phylogenetic relationships among laurasiatherian mammals at the ordinal level are still a matter of debate, the most important departures from current hypotheses detected here was the sister group relationship between Eulipotyphla and Carnivora (Fig. 1) and the sister group relationship between Pholidota and Cetartiodactyla. Specifically, in most studies, eulipotyphlan species appear sister to all other laurasiatherian mammals (Nery et al., 2012; Foley, Springer & Teeling, 2016), and Pholidota is recovered as sister to Carnivora (Meredith et al., 2011; Du Toit et al., 2014). The synteny analysis provided further support for the identity of the gremlin 2 gene lineage in this group of mammals (Fig. 2); genes found downstream were well conserved in all examined species (Fig. 2). According to our survey, most species included in this study possessed four downstream genes (RGS7, FH, KMO and OPN3) that define the identity of this genomic region (Fig. 2). Although the genes found upstream were more variable, they were to some degree more conserved in the different groups (Fig.  2). For example, in both camelid species four upstream genes (RNF2, TRMT1L, SWT1 and IVNS1ABP) were detected that were well conserved (Fig.  2). Similar results were found for sheep, goat, cow, minke whale, killer whale, and baiji (Fig. 2).

Figure 2 Patterns of conserved synteny in the genomic regions that harbor gremlin 2 genes in laurasiatherian mammals.

(A) Genomic region that harbors gremlin 2 genes. (B) Conserved synteny in the genomic region that would be the putative location of the gremlin 2 gene in ruminants and cetaceans.

Molecular rates in cetartiodactyls and pholidotans

The rate of molecular evolution, as measured here using branch lengths, was variable (Fig.  1), though the most striking result was that of the accelerated evolution of cetartiodactyls and to a lesser extent that of pholidotans (Fig. 1). To test whether the rate of gremlin 2 evolution in these groups of species is significantly higher, we performed a relative rate test (Tajima, 1989) using the software MEGA 7 (Kumar, Stecher & Tamura, 2016). We compared the rate of evolution of cetartiodactyl and pholidotan sequences using the cat sequence as a reference and the mouse sequence as the outgroup. Results of this analysis confirmed what was observed in our phylogenetic tree, i.e., the rate of evolution of cetartiodactyls and pholidotans is significantly higher than that of other laurasiatherian mammals (Table S2). This rate acceleration seems to be specific to this locus, since the same test in other members of the gene family (GREM1, CER1 and DAND5) did not reveal evidence of rate acceleration in these groups (Table S3). To further investigate the evolutionary pattern of gremlin 2 in cetartiodactyls and pholidotans, we made an amino acid alignment that included, in addition to the cetartiodactyl and pholidotan sequences, representative species of the laurasiatherian orders Perissodactyla, Carnivora, Chiroptera, and Eulipotyphla. From this, we found that there are 13 synapomorphies that define the gremlin 2 genes in cetartiodactyls (Fig. 3). Among these, we identified 11 amino acid changes and two deletions (Fig. 3). Of all of the amino acid substitutions, changes at positions 34 (Tyr to Arg), 109 (His to Pro), 131 (Thr to Ala), and 132 (Ser to Ala) represent changes affecting hydrophobicity (Fig. 3). Of the 13 synapomophies observed in cetartiodactyls, only one is shared with pholidotans (Ser165Gly) (Fig. 3). Amino acid 110 also represents a synapomorphy in pangolins however the amino acid identity is different from that of cetartiodactyls (Fig. 3).

Figure 3 An alignment of gremlin 2 amino acid sequences from laurasiatherian mammals.

Amino acid positions in bold denote the 11 amino acid synapomorphies that define the sequences of pigs and camels.

Gene copy number variation and differential retention in cetartiodactyls

Most laurasiatherian species possess a single copy of the gene with the exception of pig (Sus scrofa) that has two copies on chromosome 10 (Fig. 4). As in all examined species, in pig, one of the copies (gremlin 2-T1) was found on the 5′ side of the regulator of the G-protein signaling 7 gene (RGS7) (Fig. 2). The second copy (gremlin 2-T2) was found within the RGS7 gene, specifically between exons 13 and 14 (Fig. 4). At the amino acid level both copies differed in one amino acid (position 155); gremlin 2-T1 possessed an arginine, and gremlin 2-T2 possessed a lysine.

Figure 4 Schematic representation of the gremlin 2 syntenic region in pigs.

One of the copies (gremlin 2-T1) is located on the 5′ side of the regulator of the G-protein signaling 7 gene (RGS7) whereas the second copy (gremlin 2-T2) is located within the RGS7 gene, specifically between exons 13 and 14.

Figure 5 An evolutionary hypothesis regarding the evolution of the gremlin 2 gene in cetartiodactyl mammals.

According to this model, the last common ancestor of cetartiodactyls possessed a single copy of the gene. Species belonging to the suborders Tylopoda, the group that includes camels, alpacas, vicuñas and guanacos, and Suiformes, the group that includes pigs and peccaries, were the only groups that retained gremlin 2. According to the phyletic distribution of gremlin 2, we propose that this gene was lost in the common ancestor of ruminants, hippopotamuses, and cetaceans between 56.3 and 63.5 million of years ago as a product of a chromosomal rearrangement.

Among cetartiodactyls, we observed that gremlin 2 was differentially retained (Fig. 2). Species belonging to the suborders Tylopoda (the group that includes camels, alpacas, vicuñas, and guanacos) and Suiformes (the group that includes pigs and peccaries) were the only groups in which gremlin 2 was present (Fig. 2). In cetaceans and ruminants, gremlin 2 was not present. Thus, according to the phyletic distribution of gremlin 2 within the main groups of cetartiodactyls, the most likely scenario is that the deletion of the gene occurred between 56.3 and 63.5 million of years ago in the common ancestor of the clade that includes ruminants, hippopotamuses, and cetaceans (Fig. 5). However, until information regarding gremlin 2 in hippopotamuses is obtained, caution must be taken when interpreting these results. If, in the future, the hippopotamus genome is found to possess gremlin 2, we can determine that two independent gene losses occurred, one in the ancestor of ruminants and a second in the ancestor of cetaceans. For now, a single gene loss event is assumed.

To gain insight into the genetic mechanisms that gave rise to the deletion of gremlin 2, we compared the chromosomal location of genes found up- and downstream of gremlin 2 in human, cow, and sheep (Fig. 6). We identified a chromosomal region of approximately 12Mb, which in human was on the 5′ side of gremlin 2 (Fig. 6; pink box), while in cow and sheep it was found in a different chromosome in relation to other genes that are linked to gremlin 2 (Fig. 6; pink box). In cow, this region was moved from chromosome 16 to 28, while in sheep it was moved from chromosome 12 to 25 (Fig. 6). As a consequence of this chromosomal rearrangement, the regions that are located up- and downstream of the chromosome piece that was moved are now located together in both cow and sheep (Fig.  6). Thus, in these species, the gene that is found on the 5′ side of gremlin 2 (FMN2) was part of the chromosomal block that was moved to a different chromosome (Fig. 6) whereas the gene located on the 3′ side (RGS7) was not. From this, we suggest that one of the break points that gave rise to the chromosomal rearrangement was the chromosomal region where gremlin 2 was located (Fig. 6). Information regarding the chromosomal location of genes found up- and downstream of gremlin 2 in cetaceans would be an important piece of information not only to understand the genetic mechanisms responsible for the deletion of gremlin 2, but also to shed light on the number of gene loss events that have occurred in the clade including cetaceans, hippopotamuses and ruminants. Thus, if cetaceans and ruminants show similar patterns, we could speculate that this genetic event occurred in the last common ancestor of the group and was inherited by all descendant lineages (cetaceans, hippopotamuses and ruminants). However, if cetaceans and ruminants show different patterns, we can postulate two deletion events. Preliminary information from baijishows a similar pattern to that seen in ruminants, supporting the hypothesis that one gene loss occurred in the last common ancestor of cetaceans, hippopotamuses and ruminants (Fig. S1).

Figure 6 Schematic representation of the chromosomal regions that harbor genes located up- and downstream of gremlin 2

(A) Chromosomal region that harbors genes that are up- and downstream of gremlin 2 in humans. (B) Chromosomal regions (chrs 16 and 28) that harbor genes that are located up- and downstream of the putative location of gremlin 2 in cow. (C) Chromosomal regions (chrs 25 and 12) that harbor genes that are located up- and downstream of the putative location of gremlin 2 in sheep.

From a biomedical perspective, the loss of gremlin 2 (e.g., in cow, sheep, goat, dolphins, whales) represents a natural gene knockout (evolutionary mutant models according to Albertson et al., 2009), thus presenting an outstanding opportunity to understand gremlin 2 biology. From a physiological standpoint, this phenomenon is interesting as gremlin 2 plays a role in several developmental processes, including organogenesis, body patterning, and tissue differentiation. Thus, several questions regarding the mode of action of this gene could be investigated considering the lack of this gene in certain species. For example, what happens with BMP2 and BMP4 in the absence of gremlin 2? Are these BMPs free of any antagonist action? Or does another member of the DAN gene family fulfill gremlin 2’s molecular role? From a phenotypic perspective, it has been shown that BMP2 and BMP4 are involved in the signaling pathway that regulates tooth development (Aberg, Wozney & Thesleff, 1997; Nadiri et al., 2004). Genetic manipulation experiments have shown that gremlin 2 deficient mice have upper and lower incisor teeth with markedly reduced breadth and depth, and the upper incisors are more severely affected than the lower ones (Kantaputra et al., 2015; Vogel et al., 2015). This supports the argument that the lack of gremlin 2 contributes to the divergent dental phenotype of ruminants and cetaceans. Ruminants do not have incisors in the upper jaw; instead they have a dental pad. With the exception of elk and red deer, canines are also absent in most species. This particular dental phenotype affects how ruminants eat, which differ from phylogenetically related species that have incisors in the upper jaw (e.g., horse). For example, cows use their tongue to wrap and pull leaves into their mouths between the incisors of the lower jaw and the dental pad; thus, plants are not clearly cut during feeding. This contrasts with the feeding method of phylogenetically related species that have upper and lower incisors; these species cut plants and graze deeply. Once the food is in their mouths, cows swing their heads to chew the food slightly and mix it with saliva before swallowing. This lateral chewing action is required to cut plant tissues because molars and premolars of the maxillary jaw are wider than those located on the mandibular jaw. Conversely, sheep use their lips and teeth as their primary tools for food prehension. Their lips are used to bring food into their mouths and the incisors of the lower jaw in combination with the dental pad allow them to cut leaves. As a consequence, sheep can bite closer to the ground and have the ability to be more selective.

The loss of gremlin 2 in cetaceans is more complicated to interpret considering that one subgroup (toothed whales) has teeth while another subgroup (baleen whales) does not. To further complicate this scenario, it has been argued that it is impossible to define teeth homology between toothed whales and non-cetacean mammals (Armfield et al., 2013). From a developmental perspective, it has been demonstrated that the pathway that controls tooth differentiation and number in cetaceans is different from the typical mammalian pattern (Armfield et al., 2013). Particularly interesting is that the expression pattern of BMP4, one of the main targets of gremlin 2, differs between cetaceans and non-cetacean mammals (Sudo et al., 2004). The case of hippopotamuses remains an open question until the genome is sequenced. However, we can speculate that, as has been shown in cetaceans, the tooth morphology of this group could be related to different regulatory pathways controlling teeth development as a consequence of the absence gremlin 2.

Finally, the fact that pangolins, a group of species that do not have teeth, possess a putatively functional copy of gremlin 2 highlights that the toothless phenotype has been achieved by genetic modifications in tooth-specific genes (e.g., C4orf26; Springer et al., 2016). The presence of toothless species that possess (e.g., pangolins) and do not possess (e.g., baleen whales) gremlin 2 supports this idea. Following the same argument, and given the relationship between gremlin 2 and upper jaw incisor development (Kantaputra et al., 2015; Vogel et al., 2015), we should expect that species that lack upper jaw incisors (e.g., ruminants) but retain all other teeth should not share the modification in tooth-specific genes present in toothless species (Springer et al., 2016). Outside Laurasiatheria, armadillos could be seen as an exception to this as they do not possess incisors but they have a putatively functional gremlin 2 gene. However, it has been described that armadillos have four to six primordial incisors at birth, which means that incisors are developed during embryogenesis but are absorbed shortly after birth (Capizzo et al., 2016).

Concluding remarks

Our results show that in cetartiodatyl mammals gremlin 2 has experienced a mixture of gene loss, gene duplication, and rate acceleration. Although the last common ancestor of cetartiodactyls possessed a single copy of the gene, species belonging to the suborders Tylopoda (the group that includes camels, alpacas, vicuñas, and guanacos) and Suiformes (the group that includes pigs and peccaries) are the only groups that have retained gremlin 2 (Fig. 5). These groups also experienced acceleration in the rate of evolution of this gene, and it is this that, clearly differentiates them from all other laurasiatherians (Fig. 3). The fact that all amino acid changes that define the gremlin 2 gene in Tylopoda and Suiformes are present in both groups suggests that this gene and its corresponding protein were remodeled in the last common ancestor of cetartiodactyls and subsequently inherited by all descendant lineages (Fig. 5). After that gremlin 2 was probably lost in the ancestor of ruminants, hippopotamuses, and cetaceans between 56.3 and 63.5 million of years ago (Fig. 5). By removing a biological constraint imposed by the presence of gremlin 2, the lack of this gene could explain teeth diversity in these groups of mammals. Thus, the results presented here support the argument that gene loss is a way to increase phenotypic diversity (Olson, 1999; Albalat & Cañestro, 2016) and that gremlin 2 is a dispensable gene at least in this group of mammals.

Supplemental Information

Supplemental Information 1 Supplementary material

Click here for additional data file.

Additional Information and Declarations

Competing Interests

Author Contributions

Data Availability

The authors declare there are no competing interests.

Juan C. Opazo conceived and designed the experiments, performed the experiments, analyzed the data, contributed reagents/materials/analysis tools, wrote the paper, prepared figures and/or tables, reviewed drafts of the paper.

Kattina Zavala conceived and designed the experiments, performed the experiments, analyzed the data, prepared figures and/or tables.

Paola Krall analyzed the data, reviewed drafts of the paper.

Rodrigo A. Arias reviewed drafts of the paper.

The following information was supplied regarding data availability:

The raw data has been supplied as a Supplementary File.

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
