# Peer review of "Evolution of gremlin 2 in cetartiodactyl mammals: gene loss coincides with lack of upper jaw incisors in ruminants"

_PeerJ, doi:10.7717/peerj.2901_

## Round 0.1 · original submission · Major Revisions

Both reviewers wrote thoughtful, balanced, constructive evaluations of your paper. Your paper will be very much improved by following their advice. Please revise according to their suggestions, and resubmit the paper. Any one of the revisions that the reviewers called for is "minor" by me, but since there are so many of them, I think it is fair to say your task is to revise your paper in a "major" way.

·

Basic reporting

The figure legends should be extended to better describe the content of figures.

Experimental design

No comments

Validity of the findings

Opazo et al. have analyzed the evolutionary history of gremlin 2 in mammals, specifically in the cetartiodactyl clade that possesses characteristic teeth morphology. Based on the identification of gremlin 2 gene in genomic databases of several mammal species, combined with phylogenetic reconstructions and syntenic analyses, they have proposed that gremlin 2 gene was lost in the last common ancestor of ruminants and cetaceans, likely linked with a chromosomal rearrangement in this lineage, while the evolutionary rate of gremlin 2 gene increased in cetartiodactyl species that retained the gene.

I think that this work is a nice example of how changes in evolutionary constraints might shape differently the evolutionary history of the genes in each lineage. The work of Opazo et al. is, therefore, interesting under an evolutionary point of view for a general audience interested in the evolution of genes and gene families during animal diversification. I think that the article deserves to be published in Peerj, although some revision is required:

1. Concerning the molecular rates and structural divergence of Grem2 gene in cetartiodactyls:
• From their relative rate test, authors conclude that Grem2 gene has an accelerated evolutionary rate in cetartiodactyls. I am wondering if such acceleration it is something specific for this gene or, on the contrary, this accelerated rate is a general feature of many cetartiodactyl genes. I think that such information is relevant for evaluating evolutionary implications of the analysis.
• In addition, Grem2 is a member of the DAN gene family that comprises several BMP inhibitors, including Grem1. Again, to provide a wider perspective for the evolutionary implications, the rate of Grem1 (and other DAN genes) in the same group of species should be analyzed and commented in the article.
• In pig, there are 2 Grem2 genes. Is there any difference in the evolutionary rate of each copy? It is not unusual that after duplication, one copy evolves faster than the other copy. The localization of Grem2-T2 inside the Rgs7 gene is surprising. Is there any hypothesis about the duplication and the origin of Grem-T2?
• In absence of a functional link between the changes in the secondary structure of the Grem2 proteins and changes in the functionality of the cetartiodactyl proteins, the relevance of the three identified regions in cetartiodactyls that diverge compared to other laurasiatherian species is very limited. I suggest reconsidering this section.

2. Concerning the deletion of Grem 2 gene in ruminant and cetacean lineages, and the genetic mechanisms that gave rise to the deletion of gremlin 2:


From the comparative analysis of human, cow and sheep gene locations (Figure 7), the authors suggest that a chromosomal rearrangement might be involved in the loss of Grem2 gene in ruminants, although they cannot determine whether one ancestral or two independent losses occurred during the evolution of cetartiodactyls. I think that the extension of the gene location analysis to available cetacean species might help to discern between these two possibilities: If the chromosomal translocation found in cow and sheep is not shared by cetacean species, it would suggest two independent gene loss events, whereas if the same chromosomal rearrangement is shared by cetartiodactyl species, a single and ancestral gene loss event appears a more likely scenario.

Reviewer 2 ·

Basic reporting

Please see below

Experimental design

Please see below

Validity of the findings

Please see below

Additional comments

The authors investigated the presence/absence and molecular evolution of gremlin 2 in cetartiodactyls and other laurasiatherian mammals. Their main finding is that gremlin 2 was lost in the common ancestor of ruminants and cetaceans based on synteny analyses. They also report that rates of evolution are accelerated in this gene in camels and pigs. These conclusions are of some interest, but much of the rest of the paper is highly speculative. Specifically, the authors suggest that GREM2 constraints phenotypic evolution and that its loss in ruminants + cetaceans has promoted phenotypic variation of cetartiodactyl dentition including the loss of the incisors in ruminants. This conclusion must remain highly speculative because GREM2 is pleiotropic (as acknowledged in this paper) and it is now clear if its loss in ruminants + cetaceans is associated with increased phenotypic variation in teeth or some other function. I have some suggestions and comments below to improve the paper.

1. Dasypus novemcinctus (nine-banded armadillo) also lacks incisors, but retains a putatively functional GREM2 gene based on NCBI Gnomon prediction software and the available genome sequence (AAGV03193281). So, it would appear that GREM2 loss is not essential for getting rid of incisors.

2. Why did the authors include representatives of all laurasiatherian orders except for Pholidota (pangolins) (lines 119-129)? Pholidota is perhaps the most interesting laurasiatherian because pangolins are edentulous (toothless) and have been so since at least ~45 million years ago. The presence of a putatively functional copy of GREM2 in Manis javanica (JSZB01000348) confirms that this gene is pleiotropic and functions outside of tooth development. The occurrence of a functional GREM2 in Manis javanica also suggests that deciphering and understanding the evolutionary history of this gene in mammals, including cetartiodactyls, is more complex than just tooth development.

3. The authors should provide basic information on the GREM2 gene including gene length, length of coding sequence (including variation among taxa), the number and length of different exons. It appears that their analyses are based on a single exon, but this should be stated somewhere.

4. The legend for figure 4 should include more information on secondary structure symbols.

5. Line 121. Why were second codon positions excluded from phylogenetic analyses? This is rather unconventional.

6. The authors should make their full alignment of nucleotide coding sequences available (or perhaps I missed this somewhere).

7. The authors conclude that rates of molecular evolution are accelerated for GREM2 in cetartiodactyls (camels, pig), but have not investigated whether or not this accelerated evolution is associated with positive selection or neutral evolution. The authors should perform selection analyses to determine if positive selection has occurred on key branches (i.e., stem and crown cetartiodactyl branches) in conjunction with this rate acceleration.

Minor comments:

Line 227. "Tylopoda" instead of "tylopoda"

Line 228. "Suiformes" instead of "suiformes"

Line 125. Use "Large flying fox" instead of megabat for Pteropus vampyrus. Also make this change on line 149.

---

## Round 0.2 · accepted · Accept

You dealt effectively and positively with the reviewers' comments; I can see no barriers to immediate acceptance. Well done!